# Molecular characterization of HAMP rs10421768 gene and phenotypic expression of hepcidin; a case-control study among sickle cell anaemia patients in Ghana

**Samuel Kwasi Appiah**[1,2]\*, **Charles Nkansah**[1,2], **Gabriel Abbam**[2], **Felix Osei-Boakye**[3], **Kofi Mensah**[1,2], **Simon Bannison Bani**[4], **Solomon Chemogo**[5], **Lydia Sarpong**[4], **Takyi Godfred Addae**[4], **Daniel Boamah Sefa**[4], **Richard Adu Croffien**[4], **Larry Adom**[4], **Rekhiatu Oboirien Abdul Rauf**[4], **Farrid Boadu**[6], **Godfred Appiah Amoah**[4], **Ejike Felix Chukwurah**[1]

1 Faculty of Health Science and Technology, Department of Medical Laboratory Science, Ebonyi State University, Abakaliki, Nigeria, 2 Department of Haematology, School of Allied Health Sciences, University for Development Studies, Tamale, Ghana, 3 Faculty of Applied Science and Technology, Department of Medical Laboratory Technology, Sunyani Technical University, Sunyani, Ghana, 4 Department of Biomedical Laboratory Sciences, School of Allied Health Sciences, University for Development Studies, Tamale, Ghana, 5 Paediatric Unit, Methodist Hospital Wenchi, Wenchi, Bono Region, Ghana, 6 Seth Owusu-Agyei Medical Laboratory, Microbiology/Molecular Biology, Kintampo Health Reasearch Centre, Kintampo, Ghana

\* appiahs30@yahoo.com

**Data Availability Statement:** All relevant data are within the manuscript and its Supporting Information files.

## Abstract

### Background

The sporadic nature of blood transfusion therapy coupled with the alteration of HAMP genes may exacerbate the risk of iron burden in sickle cell anaemia (SCA) patients. The study determined the polymorphic distribution of the HAMP promoter gene rs10421768 and hepcidin levels in SCA patients.

### Method

Sixty participants aged ≥12years [45 SCA patients and 15 controls (HbA)] were recruited from 15th March, 2023 to 20th July, 2023 for a case-control study at Methodist Hospital Wenchi, Ghana. Complete blood count and hepcidin levels assessment were done using haematology analyzer and ELISA, respectively. Genomic DNA was extracted using the Qiagen Kit, and HAMP gene rs10421768 (c.-582 A>G) was sequenced using the MassARRAY method. Data were analysed using SPSS version 26.0.

### Results

The frequencies of the HAMP promoter rs10421768 genotypes AA, AG, and GG were 64.4%, 33.3%, and 2.2% in SCA patients, and 86.7%, 13.3%, and 0% in the controls, respectively. Serum hepcidin levels were significantly higher among controls than cases [204.0 (154.1–219.3) vs 150.2 (108.1–195.6)µg/L, p<0.010]. Participants with HAMP rs10421768 homozygous A genotype had higher serum levels of hepcidin compared with those in the wild genotypes (AG/GG) group [(188.7 (130.9–226.9) vs 136.8 (109.7–157.8)

**Funding:** The author(s) received no specific funding for this work.

**Competing interests:** The authors have declared that no competing interests exist.

µg/L, $p<0.016$]. Disease severity and blood cell parameters were not associated with the HAMP variants ($p>0.05$).

## Conclusion

The HAMP promoter rs10421768 AA genotype has the highest frequency of distribution and the GG genotype with the least distribution. Participants with HAMP rs10421768 G allele (c.-582A>G) had reduced levels of hepcidin. HAMP rs10421768 genotypes had no association with blood cell parameters and disease severity. The HAMP rs10421768 genotypes may influence serum levels of hepcidin. Further study is required to elucidate the potential effect of the G allele on hepcidin transcription.

## Introduction

Sickle cell anaemia (SCA) is a genetic condition involving the homozygous inheritance of haemoglobin S (HbS) with a wide spectrum of disorders [1]. It occurs as a result of a transversion mutation in the second nucleotide at the sixth position (GAG to GTG) of the beta-globin gene resulting in abnormal HbS, which has poor solubility when deoxygenated [2]. Under hypoxic conditions, haemoglobin S polymerises to form tactoids with increased red blood cell adherence to the vascular endothelium [3]. Polymerised haemoglobin S is known to be central to vaso-occlusive crisis in SCA patients and leads to secondary processes such as inflammation, haemolysis, anaemia, vasculopathy and oxidative stress affecting many organs [4,5].

The burden in sub-Saharan Africa is predicted to be 64% of the global 400,000 children born with sickle cell disease (SCD) annually with an increased death rate [6]. In Ghana, almost 2% of newborn babies have SCD, with approximately 56% having sickle cell anaemia [7].

Hydroxyurea, penicillin V and folic acid therapy have been used as the primary management protocols for SCA [8]. Blood transfusion plays a significant role in the management of patients with SCA by reducing the proportion of HbS, to limit haemolysis and endothelial damage that results from high proportions of sickle polymer-containing red cells. The blood transfusion is usually used as supportive care for patients who experience severe form of the condition and are unresponsive to the first line of medication [9,10]. Although, transfusion may treat symptoms of anaemia or prevent complications of SCD-related vaso-occlusion such as stroke and severe acute chest syndrome, it is not without drawbacks. An average healthy person's body has 4 g of iron, whereas people who receive multiple and frequent transfusions can accumulate 5 to 10 g annually. Long-term transfusion can lead to iron overload which may result in further complications including heart failure, growth retardation, endocrine problems and splenomegaly [11,12].

The body has no definite mechanism for eliminating excess iron; therefore, iron metabolism, storage and transport are tightly regulated through the hepcidin-ferroportin axis to avoid accumulation [13]. Hepcidin (major iron regulator) is a small antimicrobial peptide encoded by the hepcidin antimicrobial peptide (HAMP) gene located on chromosome 19q13. The gene is composed of three exons and has a length of 2637 bp. [14]. Research has shown that hepcidin mostly expressed by the hepatocytes is stimulated by iron overload (mediated through the bone morphogenetic protein (BMP) signalling pathway) and inflammation (by interleukin 6 (IL-6) and other cytokines through the Janus kinase/signal transducer and activator of transcription (JAK/STAT) signalling pathway), whereas erythropoietic activity, anaemia or hypoxia suppress its synthesis [15,16]. Hepcidin and ferroportin form a complex, which is

absorbed by these cells, where it causes ferroportin to degrade, inhibiting iron outflow and, as a result, reducing intestinal iron absorption and bioavailability [17]. Similar to other genes, various mutations in the HAMP gene have been reported.

It has been postulated that, the presence of G allele (c.-582A>G) may downregulate hepcidin transcription, which promotes iron absorption [16]. Previous studies reported an association between the hepcidin promoter c.-582 A>G (rs10421768) polymorphism and iron overload in patients with β-thalassaemia major [18,19]. The different transcriptional activation that happens through E-boxes inside the gene's promoter region may be the cause of the association between the HAMP gene variation and iron metabolism [19]. Another study found reduced levels of hepcidin in severely anaemic children with SCD, independent of inflammation or markers of erythropoiesis [20]. Again, a previousstudy reported that SCA patients with history of multiple blood transfusions had elevated serum hepcidin levels compared to the control groups, and the anaemia of chronic inflammation was found to be a contributing factor to the anaemia of SCD patients [21].

There is paucity of data regarding on the distribution of hepcidin (rs10421768) promoter gene polymorphisms and its expression among SCA patients in Ghana, hence, the need for this study. Findings from this study would provide the possible relationship between the HAMP polymorphic variants and hepcidin protein expression in SCA patients.

## Materials and methods

### Study site/design

This was a hospital-based case-control study conducted at Methodist Hospital, Wenchi, a Christian faith-based hospital in the Bono Region of Ghana. This study was carried out from 15th March, 2023 to 20th July, 2023. The hospital, which has 250 beds, serves as a referral hub for 20 public and private healthcare facilities in the Wenchi municipality and some parts of the Savannah region. The hospital offers specialist surgical services, including orthopaedic, urology, and general surgery. The digital address of the hospital is BW-0005-0306. The population of the municipality according to the 2021 population and housing census stands at 124,758, with 60,960 males and 63,798 females with majority of the inhabitants been farmers [22].

### Ethical consideration

All procedures were carried out in accordance with the ethical standards of the Institutional Review Board (IRB) of the University for Development Studies, Ghana (UDS/RB/006/23).

Permission was sought from the management of Methodist Hospital, Wenchi. Participants 18years and above provided written informed consent. The consents of the study participants below 18 years were obtained from guardians or parents.

### Study population

The study included 45 SCA patients, and 15 healthy (HbA) controls aged ≥12 years. The control participants were people without sickle cell disease who reside in the same geographical area and of similar ethic group as the cases. Participants below the age of 12 years with haemoglobin variants other than HbS and co-morbidities such as diabetes mellitus, hypertension, human immunodeficiency virus (HIV), and hepatitis were excluded from the study. Pregnant women, lactating mothers, and those who refused to provide consent were also excluded.

## Sample size determination

The necessary sample size was obtained by employing the Kelsey's formula:

$$N_{cases-Kelsey} = \left[\frac{r+1}{r}\right]\frac{P(1-P)\left(Z_{\frac{\alpha}{2}}+Z_{\beta}\right)^2}{(\text{p1}-\text{p2})^2}, \text{ and } \quad \text{P} = \left[\frac{p1+(r\ X\ p2)}{r+1}\right]$$

$N_{cases-Kelsey}$ is the required sample size for the sickle cell anaemia group.

r is the ratio of sickle cell anaemia group to the group without SCA, which is 2:1 in this study.

$Z_{\frac{\alpha}{2}}$ represents the critical value of the normal dispersion at α/2 (for this study at confidence level of 95%, α is 0.05 and the critical value is 1.96).

$Z_{\beta}$ represents the critical value of the normal distribution at β (this study used a power of 80%, β is 0.2 and the critical value is 0.84.

p1 represents the proportion of HAMP mutant allele in SCA group, 8.5% [23].

p2 is the proportion of HAMP mutant allele in the control group, 6.7% [23]

p1-p2 is the smallest difference in proportions that is clinically important.

The minimum number of participants with SCA patients required for this study was 30 with corresponding participants without SCA at 15.

However, this study employed 60 participants: 45 SCA patients and 15 controls (age-matched HbA healthy group).

## Data collection techniques and tools

Demographic characteristics such as age and sex were obtained from the participants. The severity of the sickle cell disease was determined using a modified method as proposed by Hedo eta al. [24], based on the clinical presentation of the patients which included the number of blood transfusions received per year, number of vaso-occlusive crises per year, number of hospitalisations per year due to relapse or crisis that require medical attention, use of hydroxy-urea, iron chelation therapy, presence of acute chest syndrome, osteomyelitis, renal failure, heart failure, avascular necrosis of the femoral hand, pneumonia, dehydration, anaemia, pigment gallstones and jaundice obtained from the clinical records. The total severity score was classified as mild SCA (Score <3), moderate SCA (Score >3 and ≤ 7) and severe SCA (Score > 7) with detailed description in previous study by Hedo et al. [24].

Categorization of blood cell transfusion was done as follows: multiple red cell transfusion (regular) when the subject received ≥ 3 units/year, rare transfusion when a client received <3 unit/year and nil transfusion when a patient received no transfusion per year.

## Sample collection and processing

Five millilitres of venous blood samples from each patient were aseptically collected and divided into two tubes: 3 mL into ethylenediamintetraacetic acid (EDTA) and 2 mL into a gel tube. The EDTA sample was used for complete blood count estimation, sickling and Hb electrophoresis tests. Genomic DNA was extracted from whole blood using the spin protocol with commercial DNA extraction kit (Qiagen, Germany), and segments of DNA encompassing the HAMP rs10421768 (c.-582 A>G) promoter gene were sequenced (Inqaba Biotec[TM] West Africa Ltd, Ghana). The gel tube sample was allowed to clot and spun at 3500 rpm for five minutes. The resulting serum was transferred into Eppendorf tubes and stored at -20˚C for hepcidin estimation using the enzyme-linked immunosorbent assay (ELISA) at University for Development Studies Laboratory.

## Haematological and biochemical determinations

The EDTA whole blood samples were used for analysis of the complete blood count using an automated five-part haematology analyser (Sysmex XN-550, Japan).

Sickle slide test was done using sodium metabisulphite and haemoglobin variants determined using cellulose acetate electrophoresis method at an alkaline pH of 8.4.

Serum hepcidin levels were assayed by the sandwich ELISA method using commercially prepared ELISA kits (Biobase, China). Microtitter plate wells were coated with purified human hepcidin antibody forming a solid phase antibody. Patients/control sera were added to the wells, incubated and upon the addition of horseradish peroxidase (HRP) labeled antibody (conjugate), antibody-antigen-enzyme-labeled antibody complex was formed. After washing using WHYM201 microplate washer and the addition of TMB substrate solution, a blue coloured product was formed. Weak sulphuric acid solution was added to terminate the reaction and the optical density of the resulting colour change was measured using ELISA reader (Powean-Medical, China) at 450 nanometers.

## DNA extraction

Genomic DNA was extracted from patients' whole blood at the Kintampo Health Research Center using the QIAGEN kits (QIAGEN, Germany), according to the manufacturer's instructions. In a 1.5 mL microcentrifuge tube, 200 μL of blood sample and 20 μL of protease K were combined. Two hundred (200) μL AL buffer was added, vortex for 15 seconds and incubated for 10 minutes at 56°C. Brief centrifugation was done at 8000 rpm to clear the drips from the lid's interior. The sample was mixed thoroughly for 15 seconds, 200 μL of ethanol was added before brief centrifugation. The mixture was put into a QIAamp Mini spin column, centrifuged for 1 minute at 8000 rpm, 500 μL of AW 1 buffer was added, centrifuged for 1 minute and residue discarded. A 500 μL of AW2 buffer was subsequently added, centrifuged at 14000 rpm for 3 minutes to eliminate the residue. After dry centrifugation, 200 μL of AE buffer added, incubated for 1 minute at ambient temperature and centrifuged for 1 minute at 8000 rpm. The concentration of DNA (μg/μL) (OD 260 x 50 ng/μL) yielded from the extraction was evaluated using a NanoDrop spectrophotometer. The extracted DNA was stored at −70°C until analysis [25].

## Identification of the HAMP gene polymorphisms

Genotyping was performed by real-time polymerase chain reaction (PCR) using the Agena MassARRAY method with iPLEX® PCR (Inqaba Biotec[TM] West Africa Ltd, Ghana).The SNP Genotyping assay consisted of an initial locus-specific PCR reaction, followed by a single-base extension using mass-modified dideoxynucleotide terminators of an oligonucleotide primer (Table 1), which annealed immediately upstream of the polymorphic site of interest. Using MALDI-TOF mass spectrometry, the distinct mass of the extended primer identifies the SNP allele as detailed in the technical manual [26]. The cluster plots for rs10421768 using the Agena MassARRAY System is shown in Fig 1.

**Table 1. Primers used for the study.**

| HAMP rs10421768 | Nucleotides |
| --- | --- |
| rs10421768_W1_F | ACGTTGGATGGTTTGAAGCTTTGGGCTACG |
| rs10421768_W1_R | ACGTTGGATGTGGAAACCCATGGAGTTCG |
| rs10421768_W1_E | TGTTCGTGTTCTATGAT |

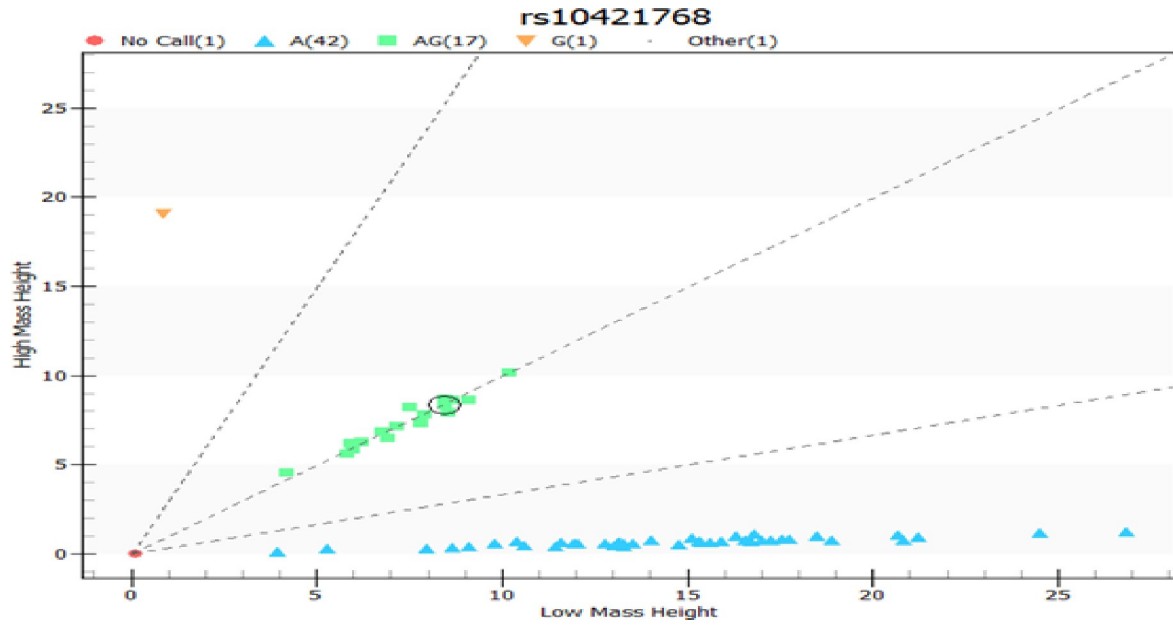

**Fig 1. Cluster plots for rs10421768 output.**

### Statistical analysis

Statistical Package for the Social Sciences (SPSS) software, version 26.0 (Armonk, NY, USA) was used for the statistical analysis. Shapiro-Wilk and one-sample Kolmogorov-Smirnov tests were used to assess the distribution of the continuous data. Parametric data were presented as mean±standard deviation whilst non-parametric data were presented as median ($25^{th}$-$75^{th}$ percentiles). Categorical data were appropriately compared using Pearson Chi-square or Fisher's exact test. Bivariate continuous data were compared using independent sample T-test (for parametric data) or Mann-Whitney test (for non-parametric data). A $p < 0.05$ was considered statistically significant.

## Results

### Demographic and allele frequency of HAMP rs10421768 gene

Table 2 shows the demographic and allele frequency of the HAMP polymorphism rs10421768 of the study participants. The 60 participants included in the study consisted of 45 (75.0%) sickle cell anaemia patients, and 15 (25.0%) healthy individuals. Of the 60 participants, 34 (56.7%) were females while 26 (43.3%) were males, with a median age of 19.0 (16.0–22.0) years. The majority (42/70.0%) of the participants had the common genotype (AA), 17 (28.3%) expressed the heterozygote genotype (AG), and 1 (1.7%) had the rare genotype (GG). Among those with SCA, the majority 29(64.4%) showed the AA genotype, 15 (33.3%) showed the AG genotype and just 1(2.2%) of the participants showed the GG genotype. In the control group, the majority 13(86.7%) expressed the common genotype (AA), with a few of them 2(13.3%) expressing the heterozygous type (AG). No significant difference was observed in the distribution of rs10421768 variants with the participants' type ($p = 0.326$) and sex ($p = 0.064$).

### Relationship between HAMP rs10421768 variants and disease severity

Table 3 displays the relationship between HAMP promoter gene variants and disease severity among the sickle cell anaemia patients. Cases with the mild form of the disease showed

**Table 2. Demographic and allele frequency of HAMP rs10421768 gene.**

| Variable | Category | HAMP rs10421768 SNP | | | Total (%) | P-value |
|---|---|---|---|---|---|---|
| | | **AA (%)** **42 (70.0)** | **AG (%)** **17 (28.3)** | **GG (%)** **1 (1.7)** | | |
| Participants | Controls (HbA) | 13 (86.7) | 2 (13.3) | 0(0.0) | 15 (25.0) | 0.326 |
| | Cases (HbS) | 29 (64.4) | 15 (33.3) | 1(2.2) | 45 (75.0) | |
| Sex | Male | 22 (84.6) | 4 (15.4) | 0 (0.0) | 26 (43.3) | 0.064 |
| | Female | 20 (58.8) | 13 (38.2) | 1(2.9) | 34 (56.7) | |
| Age | 19.0 (16.0–22.0) | | | | | |

HAMP: Hepcidin Antimicrobial Peptide; SNP: Single Nucleotide Polymorphism; SCA: Sickle cell anaemia; A: Adenine; G: Guanine; N: Number. Categorical data presented in frequencies with corresponding percentages in parenthesis. Pearson's Chi-square and Fisher's exact test were used to compare the association between two categorical data. Age was presented in median (25th-75th percentiles). P<0.05 was considered statistically significant.

genotypic frequencies of AA, AG, and GG of 16 (64.0%), 8 (32.0%), and 1 (4.0%), respectively. Among cases with moderate disease, the allele frequencies of AA and AG were observed in 9 (56.2%) and 7 (43.8%) cases, respectively, while those with severe disease showed only the mutant genotype (AA) 4 (100%). There was no significant difference between HAMP promoter gene variants and disease severity.

## Serum levels of hepcidin of study participants stratified by cases and controls

Fig 2 illustrates the hepcidin levels of the study participants stratified by cases and controls. The mean serum level of hepcidin among the study participants was 158.6 (127.8–214.8) μg/. Hepcidin levels were significantly higher among control subjects compared to their counterparts with SCA [204.0 (154.1–219.3) vs 150.2 (108.1–195.6)μg/L, $p<0.010$].

## Serum levels of hepcidin of study participants stratified by HAMP rs10421768 variants

Fig 3 shows the serum levels of hepcidin of participants stratified by HAMP rs10421768 **variants**. The mean serum hepcidin levels of HAMP mutant genotype AA were significantly higher compared to the wild genotype AG/GG [(188.7 (130.9–226.9) vs 136.8 (109.7–157.8)μg/L, p<0.016].

**Relationship between HAMP rs10421768 variants and blood cell parameters of the participants** Table 4 shows the blood cell parameters of the study participants stratified by

**Table 3. Relationship between HAMP rs10421768 variants and disease severity.**

| Disease Severity | HAMP Genotypes | | | Total N (%) | P-value |
|---|---|---|---|---|---|
| | **AA (%)** | **AG (%)** | **GG (%)** | | |
| Mild Disease | 16 (64.0%) | 8 (32.0%) | 1 (4.0%) | 25 (100.0%) | 0.448 |
| Moderately Disease | 9 (56.2%) | 7 (43.8%) | 0 (0.0) | 16 (100%) | |
| Severe Disease | 4 (100%) | 0 (0.0) | 0 (0.0) | 4 (100%) | |
| **Total** | 29 (64.4%) | 15 (33.3%) | 1 (2.2%) | 45(100%) | |

G: Guanine, A: Adenine, N: Number, Categorical information is displayed as frequencies with corresponding percentages in parentheses. Association was determined using Pearson's Chi-square. Statistical significance was set at p<0.05.

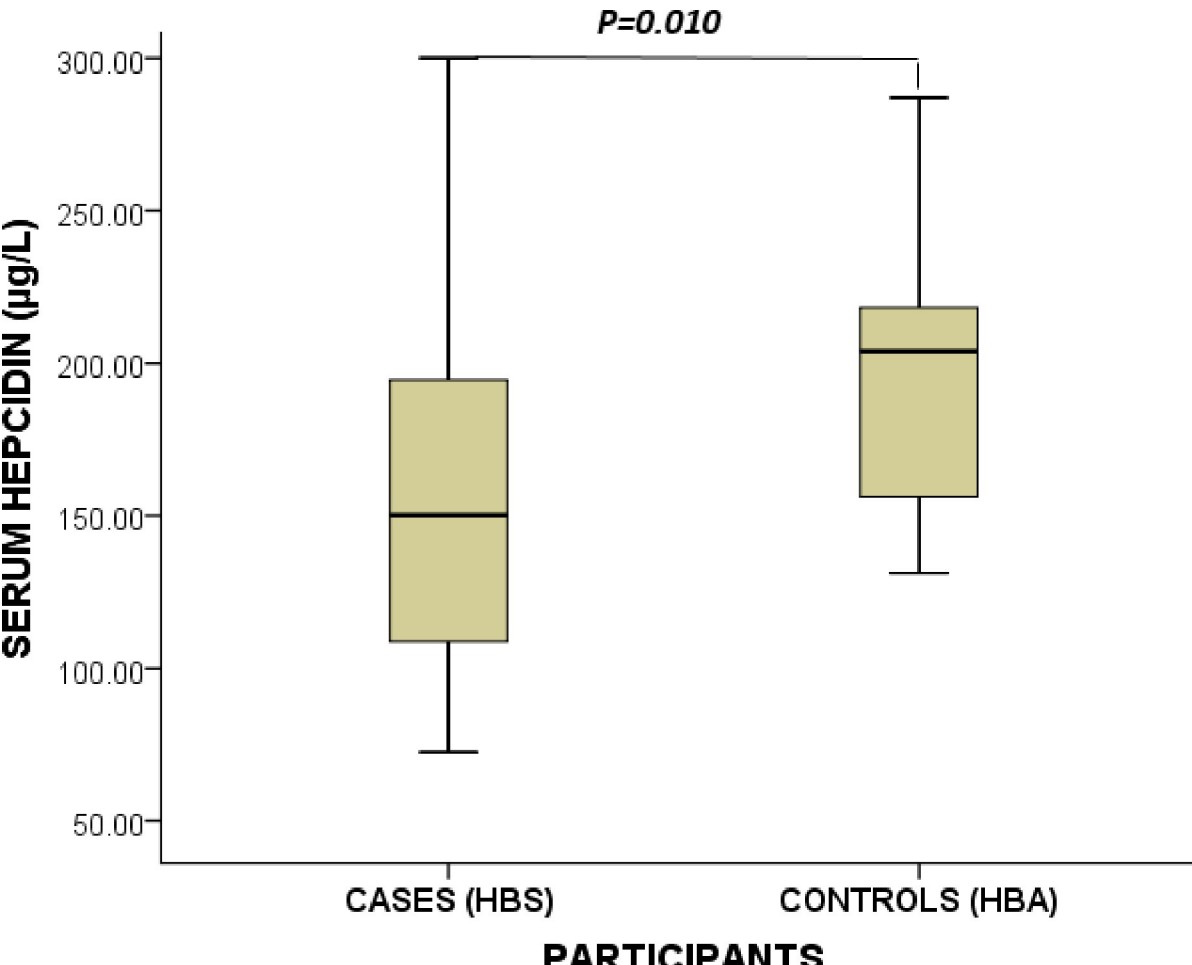

**Fig 2. Serum levels of hepcidin of study participants stratified by cases and controls.** μg/L = microgram per litre. Data were compared using the Student T-Test. P< 0.05 was deemed statistically significant.

**HAMP rs10421768** genotype variants. No statistically significant difference was observed in the blood cell parameters between the genotypes AA and AG/GG ($p>0.05$).

## Discussion

Hepcidin is essential for keeping the body's iron levels balanced, and its significance in sickle cell anaemia is vital for comprehending the intricate interplay between iron metabolism and this hereditary disease [27]. Previous studies have indicated that specific genetic variations in the HAMP promoter region can lead to diminished hepcidin expression, potentially leading to elevated serum iron levels [28–30] This study aimed to determine the genotypic variants in the hepcidin gene promoter (rs10421768) and the phenotypic levels of hepcidin protein in sickle cell anaemia patients.

In the present study, the polymorphic distribution of HAMP rs10421768 genotypes AA, AG and GG among SCA patients was 64.4%, 33.3%, and 2.2% respectively, whereas in the healthy controls, the AA, AG and GG genotypic frequencies were 86.7%,13.3% and 0% respectively. This finding corroborates with an earlier study conducted among multiple sclerosis patients that reported 75.9% frequency of A allele and 24.1% of G allele [16]. It also agrees with

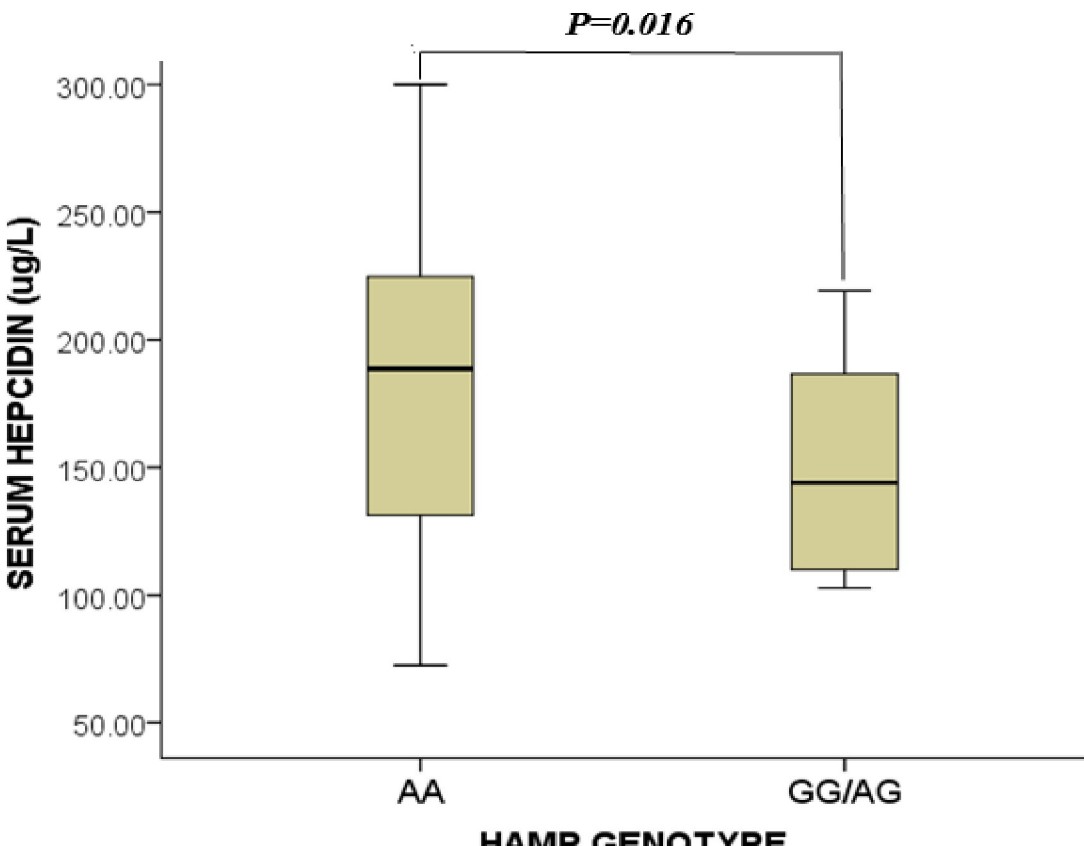

**Fig 3. Serum levels of hepcidin of study participants stratified by HAMP rs10421768 variants.** µg/L = microgram per litre. Data was compared using the Student T-Test. P<0.05 was deemed statistically significant.

**Table 4. Relationship between HAMP rs10421768 variants and blood cell parameters of the participants.**

| Blood cell parameters | HAMP rs10421768 Genotypes | | P-value |
|---|---|---|---|
| | AA N = (42) | AG/GG N = (18) | |
| RBC×10³/µL | 3.3 (2.3–4.4) | 3.0 (2.4–4.3) | 0.859 |
| Hb (g/dL) | 9.9±3.0 | 9.1±1.9 | 0.262 |
| HCT% | 29.3±8.3 | 27.2 ±5.5 | 0.270 |
| MCV (fL) | 86.0 (80.0–91.8) | 87.3 (78.6–95.1) | 0.646 |
| MCH (pg) | 30.2 (27.2–32.7) | 28.7 (26.3–31.8) | 0.388 |
| MCHC (g/dL) | 34.7 (33.3–36.0) | 34.5 (33.9–35.9) | 0.840 |
| RDW-CV% | 15.5 (9.2–18.9) | 16.9 (14.0–22.1) | 0.217 |
| TWBC×10³/uL | 7.6 (5.1–11.9) | 8.8 (7.1–11.2) | 0.287 |
| PLT.×10³/uL | 269.0 (188.3–438.0) | 270.0 (199.3–358.0) | 0.834 |

µL = Microliter; fL = Femtolitre; g/dL = Gram per deciliter; µL = microgram per litre; pg = Picrogram; N = Number of participants; RBC = Absolute red blood cell count; Hb = Haemoglobin concentration; HCT = Haematocrit; MCV = Mean cell volume; MCH = Mean cell haemoglobin; MCHC = Mean cell haemoglobin concentration; RDW-CV = Red blood cell distribution width-coefficient of variation; TWBC = Total white blood cell count; PLT. = Platelet count; Parametric data presented as mean ± standard deviation was compared using Student T-test, and non-parametric data presented as median (25th-75thpercentiles) were compared using Mann- Whitney U-test. P<0.05 was considered statistically significant.

a study by Zarghamian et al. in β-thalassemia major patients that found the frequencies of AA, AG and GG genotypes of 53.9%, 40.2% and 5.9% respectively [19].

Conversely, a previous study conducted in northern Saudi Arabia reported frequencies of HAMP promoter rs10421768 genotypes AA, AG and GG to be 3.5%, 96.5% and 0% in the iron-deficient women and 12%, 88% and 0% in the healthy women, respectively [31]. The difference in the findings may be due to the variations in the study participants and geographical locations. Whiles the current study was conducted among SCA patients in Northern Ghana, the Al-amer & Alshari study was conducted among iron deficient women in northern part of Saudi Arabia.

Additionally, the A allele was more common among males (84.6%) compared to females (58.8%), whereas the heterozygous (AG) allele was more prevalent in females (38.2%) than in males (15.4%). The homozygous G allele was rare in both sexes with only (2.9%) in females. The data obtained from the study showed an increase in A allele homozygosity of the HAMP promoter (C-582 A>G) polymorphism in those without SCA (86.7%) compared with patients with SCA (64.4%) although not statistically significant. The slight difference in genotype distribution between sexes, as indicated, may warrant further investigation to explore the potential gender-specific effects of this polymorphism on blood cell parameters and SCA susceptibility. This is highlighted by an earlier study on the genetic variation within the HAMP gene and its potential role in the context of SCA [31].

This study explored the potential link between genetic variants of HAMP rs10421768 and the disease severity in individuals with sickle cell anaemia. Even though, previous studies postulated that genetic, environmental, or clinical factors could exert considerable influence over the clinical progression and disease susceptibility [31,32], the current study did not find any association between the 582 A/G gene polymorphisms and disease severity. Notably, the majority of SCA patients, regardless of their genotype, 62.5% had mild disease, 33.3% had moderate and 8.9% had the severe form of the disease. Severe disease was uncommon in individuals with the G allele. Findings from this study corroborate a study by Parajes et al. [33] that found no association between serum iron, serum transferrin, transferrin saturation or ferritin levels and the c.-582A > G HAMP promoter variant in a healthy population [33]. The study however contradicts previous studies on potential iron overload in β-thalassaemic patients [15,18]. The variation in the findings may be attributed to demographics and differences of the study subjects. This study recruited SCA patients whereas the earlier studies recruited β-thalassaemic patients.

This study found significantly lower hepcidin levels in SCA patients than the healthy controls. Previous studies have reported similar findings and this could be explained by the overriding effect of intense erythropoiesis or hypoxia that downregulates hepcidin synthesis as against the stimulatory effect of inflammatory cytokines (IL-6) on hepcidin experienced by SCA patients [17,26]. This contradicts an earlier study in Egypt that reported elevated serum hepcidin levels in sickle cell disease subjects with multiple blood transfusions [21]. The difference in the findings may be due to the variations in the study participants. Ismail et al. [21] recruited β-thalassaemia major patients who are transfusion dependent whilst the current study recruited SCA patients.

Additionally, this study found significantly reduced serum hepcidin levels in study participants with the G allele compared to those with the A allele. This finding corroborates with previous studies [18,19,30]. The reason for the finding could be attributed to the inhibitory effect of the G allele on hepcidin transcription. The HAMP c.-582 A>G variants located in the E-box 1 acts as a responsive element for upstream stimulatory factors 1 and 2 (USF 1&2) necessary for promoting adequate transcription and synthesis of hepcidin. The presence of G

variants affects the binding of the transcription factor to the E-box which down-regulates the transcription of the HAMP gene and its expression [16].

The relationship between blood cell parameters and the HAMP rs10472168 (c.-582 A>G) were assessed. The study did not find any relationship between blood cell parameters and HAMP gene variants, which corroborates findings from earlier studies [31,34,35].

This study could not assess the iron status of the study participants.

## Conclusion

The HAMP promoter rs10421768 AA genotype has the highest frequency of distribution and the GG genotype with the least distribution. Participants with HAMP rs10421768 G allele (c.-582A>G) had reduced levels of hepcidin. HAMP rs10421768 genotypes had no association with blood cell parameters and disease severity. The HAMP rs10421768 genotypes may influence serum levels of hepcidin. Further study with large sample size is required to elucidate the potential effect of the G allele on hepcidin transcription.

## Supporting information

**S1 Data.**
(SAV)

## Acknowledgments

Authors appreciate the enormous contributions of management and staff of Kintampo Health Research Center for providing us with necessary resources for the DNA extraction. We also appreciate the immense support of University for Development Studies and the management of Methodist Hospital Wenchi, Ghana. Lastly, we thank all participants in the study.

## Author Contributions

**Conceptualization:** Samuel Kwasi Appiah, Simon Bannison Bani, Lydia Sarpong, Takyi Godfred Addae, Daniel Boamah Sefa, Richard Adu Croffien, Ejike Felix Chukwurah.

**Data curation:** Samuel Kwasi Appiah, Charles Nkansah, Gabriel Abbam, Felix Osei-Boakye, Solomon Chemogo, Daniel Boamah Sefa, Larry Adom, Rekhiatu Oboirien Abdul Rauf.

**Formal analysis:** Samuel Kwasi Appiah, Charles Nkansah, Felix Osei-Boakye, Lydia Sarpong, Takyi Godfred Addae, Richard Adu Croffien, Farrid Boadu, Godfred Appiah Amoah.

**Funding acquisition:** Samuel Kwasi Appiah, Lydia Sarpong, Takyi Godfred Addae, Daniel Boamah Sefa, Richard Adu Croffien.

**Investigation:** Samuel Kwasi Appiah, Charles Nkansah, Kofi Mensah, Solomon Chemogo, Lydia Sarpong, Takyi Godfred Addae, Daniel Boamah Sefa, Richard Adu Croffien, Larry Adom, Rekhiatu Oboirien Abdul Rauf, Farrid Boadu, Godfred Appiah Amoah.

**Methodology:** Samuel Kwasi Appiah, Felix Osei-Boakye, Kofi Mensah, Solomon Chemogo, Lydia Sarpong, Takyi Godfred Addae, Daniel Boamah Sefa, Larry Adom, Rekhiatu Oboirien Abdul Rauf, Farrid Boadu, Godfred Appiah Amoah, Ejike Felix Chukwurah.

**Project administration:** Samuel Kwasi Appiah, Charles Nkansah, Kofi Mensah, Simon Bannison Bani, Lydia Sarpong, Takyi Godfred Addae, Ejike Felix Chukwurah.

**Resources:** Samuel Kwasi Appiah, Felix Osei-Boakye, Lydia Sarpong, Takyi Godfred Addae, Daniel Boamah Sefa, Richard Adu Croffien, Farrid Boadu.

**Software:** Gabriel Abbam.

**Supervision:** Samuel Kwasi Appiah, Simon Bannison Bani, Ejike Felix Chukwurah.

**Validation:** Samuel Kwasi Appiah, Charles Nkansah, Gabriel Abbam, Felix Osei-Boakye, Kofi Mensah, Simon Bannison Bani, Ejike Felix Chukwurah.

**Visualization:** Samuel Kwasi Appiah, Charles Nkansah, Gabriel Abbam, Ejike Felix Chukwurah.

**Writing – original draft:** Samuel Kwasi Appiah, Lydia Sarpong.

**Writing – review & editing:** Samuel Kwasi Appiah, Charles Nkansah, Gabriel Abbam, Felix Osei-Boakye, Kofi Mensah, Simon Bannison Bani, Solomon Chemogo, Lydia Sarpong, Takyi Godfred Addae, Daniel Boamah Sefa, Richard Adu Croffien, Larry Adom, Rekhiatu Oboirien Abdul Rauf, Farrid Boadu, Godfred Appiah Amoah, Ejike Felix Chukwurah.

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
