## [Decision Letter · Decision Letter 0]

6 Mar 2024

PONE-D-24-00585Molecular characterization of HAMP rs10421768 gene and phenotypic expression of hepcidin; a case-control study among sickle cell anaemia patients in GhanaPLOS ONE

Dear Dr. Appiah,

Thank you for submitting your manuscript to PLOS ONE. After careful consideration, we feel that it has merit but does not fully meet PLOS ONE’s publication criteria as it currently stands. Therefore, we invite you to submit a revised version of the manuscript that addresses the points raised during the review process. Based on the reviewers suggestions, the paper needs major revision. The reviewers comments can be found below. Please submit your revised manuscript by Apr 20 2024 11:59PM. If you will need more time than this to complete your revisions, please reply to this message or contact the journal office at plosone@plos.org. Please include the following items when submitting your revised manuscript:A rebuttal letter that responds to each point raised by the academic editor and reviewer(s). You should upload this letter as a separate file labeled 'Response to Reviewers'.A marked-up copy of your manuscript that highlights changes made to the original version. You should upload this as a separate file labeled 'Revised Manuscript with Track Changes'.An unmarked version of your revised paper without tracked changes. You should upload this as a separate file labeled 'Manuscript'.

We look forward to receiving your revised manuscript.

Kind regards,

Tanja Grubić Kezele, Ph.D., M.D.

Academic Editor

PLOS ONE

Journal Requirements:

Whilst you may use any professional scientific editing service of your choice, PLOS has partnered with both American Journal Experts (AJE) and Editage to provide discounted services to PLOS authors. Both organizations have experience helping authors meet PLOS guidelines and can provide language editing, translation, manuscript formatting, and figure formatting to ensure your manuscript meets our submission guidelines. To take advantage of our partnership with AJE, visit the AJE website (http://aje.com/go/plos) for a 15% discount off AJE services. To take advantage of our partnership with Editage, visit the Editage website (www.editage.com) and enter referral code PLOSEDIT for a 15% discount off Editage services. If the PLOS editorial team finds any language issues in text that either AJE or Editage has edited, the service provider will re-edit the text for free.

3. We are unable to open your Supporting Information file HAMP DATA.sav. Please kindly revise as necessary and re-upload.

Reviewers' comments:

Reviewer's Responses to Questions

**Comments to the Author**

1. Is the manuscript technically sound, and do the data support the conclusions?

Reviewer #1: Partly

Reviewer #2: Yes

2. Has the statistical analysis been performed appropriately and rigorously? 

Reviewer #1: Yes

Reviewer #2: Yes

3. Have the authors made all data underlying the findings in their manuscript fully available?

Reviewer #1: Yes

Reviewer #2: Yes

4. Is the manuscript presented in an intelligible fashion and written in standard English?

Reviewer #1: Yes

Reviewer #2: Yes

5. Review Comments to the Author

Reviewer #1: The sample size is very limited, and the control cases are not enough for a comparable controlled study .

More cases are required for proper statistical analysis . Not using real-time PCR for SNP analysis (which the most common method in most of the similar researches ) are not justified. the discussion part could be enriched with other researches of similar population for better comparability.

The consent for the patients methodology is not clear

Reviewer #2: The study determined the polymorphic distribution of the HAMP promoter gene rs10421768 and hepcidin levels in SCA patients.

The article is presented in an intelligible fashion.

The study presents the results of primary scientific research.

Experiments, statistics, and other analyses are described in sufficient detail.

The research seems to meet all applicable standards for the ethics of experimentation and research integrity.

Conclusions are not entirely supported by the data.

The relevance of the study lies in determining whether this polymorphism implies a higher risk of developing iron overload due to decreased hepcidin synthesis. Therefore, it would be very enriching if the authors conducted a measure of iron overload, such as measuring serum ferritin concentrations, for instance.

It is important to divide the participants according to the different polymorphisms and compare the serum hepcidin concentrations to verify whether the polymorphism is indeed determining the hepcidin levels.

Considering that the study did not establish any relationship between ferritin concentrations and the mapped polymorphisms, and that the objective was to evaluate the distribution of the polymorphisms, in this case, it is essential to present a sample size calculation.

Lines 102-105: It doesn't seem appropriate to claim that the data from this study can provide appropriate guidance in managing heavily transfused individuals. I think it's important to downplay the significance. I believe the data can support future investigations. However, based on the applied methods and investigated variables, it's unclear whether the polymorphism would pose a higher risk of iron overload.

Line 139: It is important to describe the criteria used to determine the severity of the disease.

Table 2: The ethnic-racial information of the participants was not provided. Please, provide it.

Considering that the study's objective is to present the distribution of polymorphisms, it is important to have a more detailed description of the control group. Who are these participants? Are they from the same geographical region? Do they have similar socio-economic conditions? Why is the polymorphism distribution different in the control group? Please provide this information and discuss it.

Table 4: Please, you should exclude the written representation of microliter, etc

Lines 315-316: This comparison doesn't make much sense because the physiopathologies of the two diseases are completely different.

Lines 327-328: Iron overload increases hepcidin. It is not possible to compare with studies that do not separate individuals based on the presence or absence of iron overload. Or at least, a consideration should be made regarding this aspect.

Lines 337-339: please revise this sentence “…and the possible iron overload”.

The discussion should address the limitations of the study.

6. PLOS authors have the option to publish the peer review history of their article (what does this mean?). If published, this will include your full peer review and any attached files.

Reviewer #1: No

Reviewer #2: No

---

## [Author Response · Author response to Decision Letter 0]

20 Mar 2024

Dear Editor,

Thank you for your email. I am pleased to resubmit manuscript titled “Molecular characterization of HAMP rs10421768 gene and phenotypic expression of hepcidin; a case-control study among sickle cell anaemia patients in Ghana” for your consideration.

Authors do appreciate your endless efforts and rich comments to ensure improvement in the manuscript.

The concerns raised by the editorial team have been addressed and highlighted below and marked in the respective sections in the manuscript. I look forward to your favourable response.

Thank you.

Reviewers' Comments:

Reviewer #1: 

i. The sample size is very limited, and the control cases are not enough for a comparable controlled study. More cases are required for proper statistical analysis.

Response: This was part of student academic thesis without any external financial support making it difficult to recruit larger sample size. This has been duly acknowledged as limitation

ii. Not using real-time PCR for SNP analysis (which the most common method in most of the similar researches) are not justified.

Response: Genotyping was performed by real-time polymerase chain reaction (PCR) using the Agena MassARRAY method with iPLEX® PCR (Inqaba BiotecTM West Africa Ltd, Ghana) as stated in trhe methodology (line 187-192)

iii. The consent for the patients methodology is not clear.

Response: This has been revised for clarity and it reads “Participants 18years and above provided written informed consent. The consents of the study participants below 18years were obtained from guardians or parents“

Reviewer #2: 

i. Conclusions are not entirely supported by the data.

The relevance of the study lies in determining whether this polymorphism implies a higher risk of developing iron overload due to decreased hepcidin synthesis. Therefore, it would be very enriching if the authors conducted a measure of iron overload, such as measuring serum ferritin concentrations, for instance.

Response: The authors acknowledge your concerns. However, the study could not assess the iron status/profile of the participants, and this has been acknowledged as a limitation of the study 

ii. It is important to divide the participants according to the different polymorphisms and compare the serum hepcidin concentrations to verify whether the polymorphism is indeed determining the hepcidin levels.

Response: This information is presented in figure 3. Only one participant had homozygous GG variant making it statistically inappropriate to separate it.

iii. Considering that the study did not establish any relationship between ferritin concentrations and the mapped polymorphisms, and that the objective was to evaluate the distribution of the polymorphisms, in this case, it is essential to present a sample size calculation.

Response: Determination of sample size is well presented in the methodology (Line 132-148)

iv. Lines 102-105: It doesn't seem appropriate to claim that the data from this study can provide appropriate guidance in managing heavily transfused individuals. I think it's important to downplay the significance. I believe the data can support future investigations. However, based on the applied methods and investigated variables, it's unclear whether the polymorphism would pose a higher risk of iron overload.

Response: this section has been revised “findings from this study would provide the possible relationship between the HAMP polymorphic variants and hepcidin protein expression in SCA patients”

v. Line 139: It is important to describe the criteria used to determine the severity of the disease.

Response: The detailed description of the severity score has been described in previous study by Hedo et al., 1993, and this has been referenced accordingly in the manuscript.

vi. Table 2: The ethnic-racial information of the participants was not provided. Please, provide it.

Response: Data regarding ethnicity were not captured. However, the study was conducted in Akan dominated population in Ghana 

vii. Considering that the study's objective is to present the distribution of polymorphisms, it is important to have a more detailed description of the control group. Who are these participants? Are they from the same geographical region? Do they have similar socio-economic conditions? Why is the polymorphism distribution different in the control group? Please provide this information and discuss it.

Response: The control participants were people without sickle cell disease who reside in the same geographical area and of similar ethic group as the cases.

viii. Table 4: Please, you should exclude the written representation of microliter, etc

Response: this has been revised accordingly

ix. Lines 315-316: This comparison doesn't make much sense because the physiopathologies of the two diseases are completely different.

Response: This comparison has been expunged from the discussion

x. Lines 327-328: Iron overload increases hepcidin. It is not possible to compare with studies that do not separate individuals based on the presence or absence of iron overload. Or at least, a consideration should be made regarding this aspect.

Response: the comparison is on hepcidin levels and not iron. It is true iron overload increases hepcidin levels. It expression can also be inhibited by intense erythropoietic stress or hypoxia. 

xi. Lines 337-339: please revise this sentence “…and the possible iron overload”.

The discussion should address the limitations of the study.

Response: The phrase “…and the possible iro

---

## [Decision Letter · Decision Letter 1]

13 Jun 2024

Molecular characterization of HAMP rs10421768 gene and phenotypic expression of hepcidin: a case-control study among sickle cell anaemia patients in Ghana

PONE-D-24-00585R1

Dear Dr. Appiah,

We’re pleased to inform you that your manuscript has been judged scientifically suitable for publication and will be formally accepted for publication once it meets all outstanding technical requirements.

Kind regards,

Kostas Pantopoulos, PhD

Academic Editor

PLOS ONE

Additional Editor Comments (optional):

Reviewers' comments:

Reviewer's Responses to Questions

**Comments to the Author**

1. If the authors have adequately addressed your comments raised in a previous round of review and you feel that this manuscript is now acceptable for publication, you may indicate that here to bypass the “Comments to the Author” section, enter your conflict of interest statement in the “Confidential to Editor” section, and submit your "Accept" recommendation.

Reviewer #2: All comments have been addressed

2. Is the manuscript technically sound, and do the data support the conclusions?

Reviewer #2: Yes

3. Has the statistical analysis been performed appropriately and rigorously? 

Reviewer #2: Yes

4. Have the authors made all data underlying the findings in their manuscript fully available?

Reviewer #2: Yes

5. Is the manuscript presented in an intelligible fashion and written in standard English?

Reviewer #2: Yes

6. Review Comments to the Author

Reviewer #2: The authors made an effort to address all the suggestions made. I consider that the changes were sufficient, even though it was not possible to present the data on serum ferritin, which undoubtedly would bring greater relevance to the study.

7. PLOS authors have the option to publish the peer review history of their article (what does this mean?). If published, this will include your full peer review and any attached files.

Reviewer #2: No

---

## [Editor Report · Acceptance letter]

18 Jun 2024

PONE-D-24-00585R1 

PLOS ONE

Dear Dr. Appiah, 

I'm pleased to inform you that your manuscript has been deemed suitable for publication in PLOS ONE. Congratulations! Your manuscript is now being handed over to our production team.

Kind regards, 

on behalf of

Dr. Kostas Pantopoulos 

Academic Editor

PLOS ONE